# Design of Two-Degree-of-Freedom Fractional-Order Internal Model Control Algorithm for Pneumatic Control Valves

**Min Zhu** [1,2,*], **Siyuan Chen** [1], **Zihao Xu** [1] **and Xueping Dong** [1,2]

1  School of Electrical Engineering and Automation, Hefei University of Technology, Hefei 230009, China; 2022170439@mail.hfut.edu.cn (S.C.); 2020110452@mail.hfut.edu.cn (Z.X.); hfdxp@hfut.edu.cn (X.D.)
2  Engineering Technology Research Center of Industrial Automation, Hefei 230009, China
*  Correspondence: zhumin@hfut.edu.cn

**Abstract:** In response to the problems of the inaccurate pneumatic control valve model, the slow valve position control, and the low precision in the industrial control process, some improvement methods are proposed. Firstly, the fractional-order concept is introduced based on the first-order inertia model and IBBO (improved biogeography-based optimization) is used for iteration to obtain a specific transfer function model. Secondly, a fractional-order and two-degree-of-freedom combined internal model control algorithm is proposed. Finally, semi-physical experiments are carried out on a semi-physical experimental platform. The results show that in the field of pneumatic regulating valves, the fractional-order model has good adaptability and effectiveness, and the two-degree-of-freedom fractional-order internal model control algorithm also effectively improves the accuracy, speed, and robustness of the valve position control.

**Keywords:** pneumatic control valve; system identification; fractional-order; internal mode control



## 1. Introduction

The control valve, as a control terminal actuator in a smart plant, is the control part of the fluid transfer system [1], which has functions related to cutting off, regulating, grading, preventing backflow, regulating pressure, grading, or eliminating overflow pressure [2]. Pneumatic control valves inevitably have non-linear characteristics such as hysteresis and dead zones due to their sealing performance, friction, and flow characteristic curves [3]. In the process of industrial production, if the valve position is not properly controlled so that the oscillation is too large, it will increase the wear and tear of the valve stem, which can cause severe shock and reduce the life of the control valve. If the adjustment time is too long, this is not conducive to production efficiency. Pneumatic control valves not only need to quickly and smoothly reach the specified valve position but also need to have a high degree of accuracy. The performance of a control valve includes not only the design and selection of the hardware system but also the control algorithm inside the valve positioner, which often plays a critical role [4]. The hardware adjustment and update of the control valve and the innovation and improvement of the control algorithm are inseparable from the mechanism modeling and simulation experiments, based on which various scholars, enterprises, and universities at home and abroad have made significant contributions.

Sherear [5] created a linearized mathematical model of the cylinder midpoint position and obtained a differential equation with unknown parameters. Martin and McCloy [6] similarly explored the model function to Sherear's model via extrapolation, but the model was not rigorously experimentally validated and the relevant parameters were not specified. Wang [7] mathematically modeled the pneumatic PCM (pulse code modulation) position system using the positioning discrimination modeling method based on the knowledge of linear system theory. Aziz and Bone [8] described a novel automatic adjustment method for the precise position control of pneumatic actuators, which combines a model-based offline analysis and online iterations. Zhou, Shen, Tamura, Nakazawa, and Henmi [9] proposed an

adaptive nonlinear switching-type robust control strategy to adjust the valve position in a closed-loop manner, and showed experimentally that adaptive nonlinear control is effective in reducing friction and discharge fluctuations and ensures good performance in the presence of unknown plant parameters. Nguyen, Leavitt, Jabbari, and Bobrow [10] used sliding film control for pneumatic systems to extend the valve life and provide good tracking and relatively low steady-state position errors. Zhu, Ma, and Schock [11] developed an iterative model-based predictive control scheme for the control of an electric–pneumatic valve actuator (EPVA) for exhausts. Xu [12] published two strategic methods of variable forward PWM duty cycles, linearization and segmented. Lu [13] used experimental calculations and technical identification to construct a working mechanism model of intelligent valve positioner with nonlinear characteristics, and proposed the use of a Bang-Bang/PID segmented controller with an inverse gap compensation algorithm to eliminate the gap characteristics, which can finally make the controller have a good output effect and control quality. Fan [14] described a numerical simulation model of a solenoid valve and improved and optimized it so that its overall performance was improved. Wang [15] explored the effects of the valve vibration amplitude, period, frequency, and velocity on transient injection characteristics and developed a transient computational fluid dynamics (CFD) model of a gas fuel injection device, whose results showed that there is a linear relationship between the transient mass flow rate and the transient lift during the vibration process. Zhang [16] proposed an air pressure control method similar to PWM (SPWM). By controlling the opening and closing time of the solenoid valve, the brake air pressure can be precisely adjusted to improve the dynamic response characteristics of the system. Xu [17] proposed a valve opening control scheme based on variable universe fuzzy auto-disturbance rejection. The simulation results showed that the variable universe fuzzy auto-disturbance rejection controller has strong anti-interference ability and good adaptability, can quickly and stably reach the preset control opening, and can achieve precise and stable control of the valve opening.

The standard pneumatic control valve model is not accurate enough because it does not take into account the fluctuation of the gas source air pressure, system sticking, and dead zones. Although some forward-looking work has been done for the pneumatic control valve, the current pneumatic control valve still has the problems of inaccurate valve position control, a considerable amount of overshoot, and a long adjustment time, so this paper uses an improved biogeographic optimization algorithm to fit the control system open-loop response curve and derive a new pneumatic control valve model. In a previous publication [18], the authors proposed an improved biogeography-based optimization algorithm with improvements including the chaotic initialization of populations, tuning of migration models, and updating of migration operators and variation operators. In addition, the two-degree-of-freedom fractional-order internal mode control method was applied to the valve position control of the pneumatic control valve, and the effectiveness of the proposed control valve position control method was demonstrated using a simulation and experiment.

## 2. Overview of the Improved Biogeography Optimization Algorithm with an Internal Mode Control Algorithm

### 2.1. Overview of Biogeography Optimization Algorithms

Dan Simon [19] formally presented the biogeography-based optimization (BBO) algorithm at IEEE Transactions on Evolutionary Computation in 2008. The method uses the principles of biogeography for mathematical modeling and simulates the migration movements and information exchange of species between island habitats, resulting in a biogeographic optimization process. Because the BBO algorithm is simple, easy to implement, and has fewer parameters, it received a lot of attention from scholars in various countries once it was proposed [20–23].

The standard biogeography optimization algorithm contains three main operators, the migration operator, the variation operator, and the removal operator:

(1) The migration operator is used to exchange information between the selected island habitats and the roulette-selected island habitats in the remaining island habitats;

(2) (The variation operator performs random variation in one dimension of the selected island habitat within the upper and lower defined limits;

(3) The removal operator removes duplicate species from the island habitat after a series of operations on the species in the island habitat (one of the duplicate species is retained and the others are randomly mutated within the upper and lower limits).

Regarding the improvements, the main ones are the chaos initialization, migration model, migration operator, and variational operator improvements, which can be found in the author's article published in Sensors.

### 2.2. Overview of the Internal Mode Control Algorithm

In 1982, Garcia and Morari presented and developed the internal mode control algorithm in its entirety [24], as shown in Figure 1.

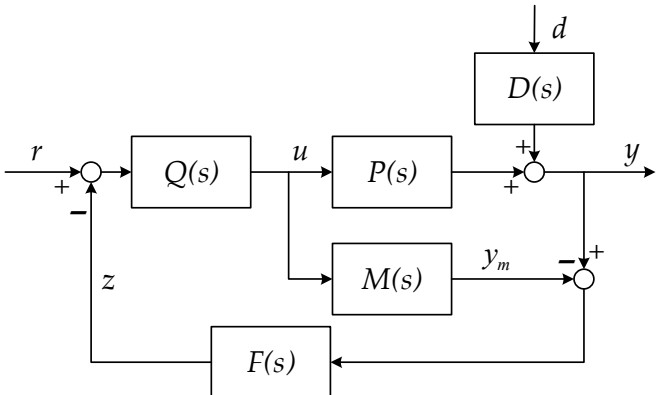

**Figure 1.** Typical internal mode control structure diagram.

For the convenience of finding the input–output relationship, let $F(s) = I$, which can be equivalently transformed from Figure 1 to the classical feedback control form as shown in Figure 2.

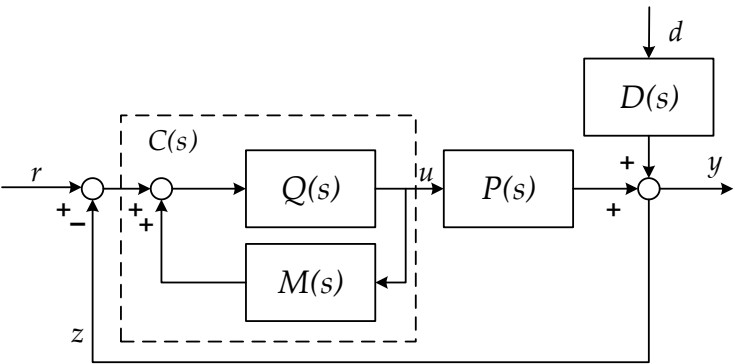

**Figure 2.** Equivalent feedback control system structure diagram.

Here, $P(s)$ is the controlled process of the control object; $M(s)$ is the mathematical model of the control object process, i.e., the so-called internal model; $Q(s)$ is the internal mode controller of the whole closed-loop system; $r$, $y$, and $d$ are the input signal, output quantity, and disturbance source of the closed-loop internal mode control system, respectively, whose control index is to maintain the stable output of $y$ while maintaining the approximation to $r$ and finally achieve the desired goal of $y = r$; $D(s)$ denotes the influence of the disturbance source on the output. In Figure 1, $F(s)$ is the feedback filter; when $F(s) = I$, the system is a single degree of freedom system, otherwise it is called a

two-degree-of-freedom system. The purpose of adding the feedback filter is to maintain the balance and improve the control effect and quality, so that the dynamic response performance and robust stability complement each other and do not conflict with each other, meaning both can ensure that the control requirements meet with the objectives [25]. In addition, the part $C(s)$ framed by the dashed line is the feedback controller, denoted as:

$$C(s) = \frac{Q(s)}{1 - Q(s)M(s)} \tag{1}$$

The following representation of the input–output relationship can be derived from Figure 2:

$$\frac{y}{r} = \frac{C(s)P(s)}{1 + C(s)P(s)}, \tag{2}$$

$$\frac{y}{d} = \frac{D(s)}{1 + C(s)P(s)}. \tag{3}$$

Substituting Equation (1) into Equations (2) and (3) yields:

$$\frac{y}{r} = \frac{Q(s)P(s)}{1 + Q(s)[P(s) - M(s)]}, \tag{4}$$

$$\frac{y}{d} = \frac{[1 - Q(s)M(s)]D(s)}{1 + Q(s)[P(s) - M(s)]}. \tag{5}$$

Thus, the closed-loop response and feedback signals of the system shown in Figure 2 are:

$$y = \frac{Q(s)P(s)}{1 + Q(s)[P(s) - M(s)]}r + \frac{[1 - Q(s)M(s)]D(s)}{1 + Q(s)[P(s) - M(s)]}d, \tag{6}$$

$$z = [P(s) - M(s)]u + D(s)d. \tag{7}$$

Equations (1)–(7) are taken from [21].

## 3. Identification

After years of development, many scholars have proposed different methods for this key problem of identification, including classical identification methods such as least squares, maximum likelihood estimation, and correlation analyses, as well as modern identification methods, especially for nonlinear systems such as setter system identification and multilayer recursive system identification. For the identification of the pneumatic control valve in this experiment, due to the high complexity of the system itself and the accuracy requirements of the identified model, the authors mainly list the least-squares-based MATLAB identification toolbox method and the identification method based on the intelligent optimization iterative algorithm in the following sections [26].

### 3.1. MATLAB Toolbox Recognition

The least squares method is used to find the best function match for the data by minimizing the sum of the squares of the errors [27]. The specific steps of using the MATLAB(2022) identification toolbox are as follows: identification data acquisition, identification data import, identification results generation, and the analysis. The raw data are acquired mainly from the system's open-loop experiments, and the normalized data of the pneumatic regulating valve are obtained. Then, this integer-order recognition result transfer function model is:

$$M(s) = \frac{K}{Ts + 1}e^{-Ls} = \frac{1.0105}{28.8610s + 1}e^{-0.8s}. \tag{8}$$

### 3.2. Intelligent Optimization of Iterative Algorithm Recognition

The intelligent optimization algorithm is used to optimize the identification of the pneumatic control valve iteratively to seek a good and excellent transfer function model that is more in line with the actual situation. In each iteration, the step response of the current model is compared with the normalized experimental data, the square of the difference is used as the fitness value of the algorithm, and the fitness value function is selected to measure the effect of the optimization search, and the fitness value function is:

$$E = \sum_{i=1}^{n} e(i)^2, \tag{9}$$

where *n* is the total number of sampling points and *e(i)* is the deviation of the predicted value from the actual value at sampling point *i*.

The flowchart of the identification iterative optimization search is shown in Figure 3.

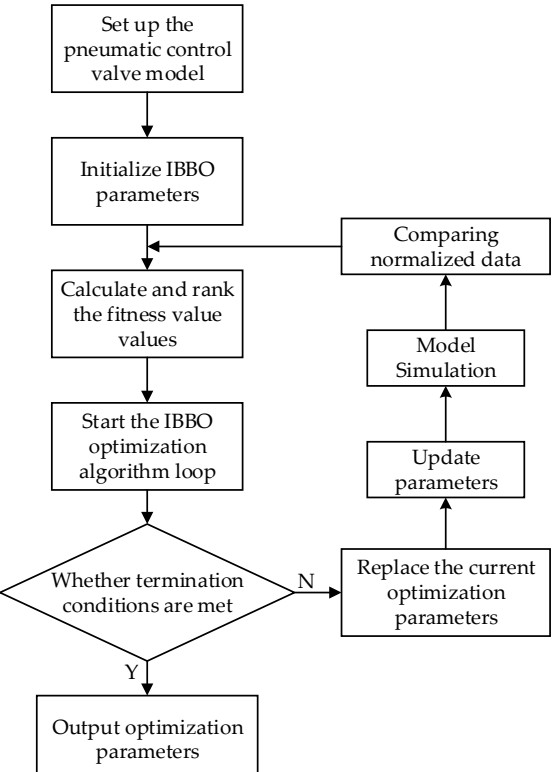

**Figure 3.** Identifying the iterative merit search flow chart.

Here, we set the algorithm population size as $M = 200$, the maximum number of iterations as $Maxgen = 200$, and the first-order time lag model parameters take a range of values of $K \in [0.0001, 100]$, $T \in [0.0001, 100]$, and $L = 0.8$. After several iterations, the pneumatic control valve integer-order transfer function model is:

$$M(s) = \frac{K}{Ts+1}e^{-Ls} = \frac{1.0127}{29.9656s+1}e^{-0.8s}. \tag{10}$$

### 3.3. Fractional-Order Inertia Model Building

If an unknown parameter of the dimension is added to the pole position of the transfer function, the first-order time lag model becomes a fractional-order time lag model, i.e.,:

$$M(s) = \frac{Ke^{-Ls}}{Ts^\lambda + 1}. \tag{11}$$

Fractional calculus involves the extension, study, and application of differential and integral operators of a non-integer-order based on classical calculus. Due to the clearer physical meaning and more accurate physical characteristics of actual systems or nonlinear systems described by fractional calculus equations, it has attracted the attention of many scholars.

At this point, we add another one-dimensional variable $\lambda \in [0.0001, 2.0000]$ and iterate this model with IBBO optimization, resulting in a fractional-order transfer function model for pneumatic control valves as:

$$M(s) = \frac{0.9905}{38.0346s^{1.0708} + 1}e^{-0.8s}. \tag{12}$$

*3.4. Model Error Analysis*

Three pneumatic control valve transfer function models are obtained in this paper:

(1) Integer-order transfer function models identified by the MATLAB identification toolbox, based on a least squares implementation:

$$M(s) = \frac{1.0105}{28.8610s + 1}e^{-0.8s}. \tag{13}$$

(2) An IBBO optimized iterative integer-order transfer function model:

$$M(s) = \frac{1.0127}{29.9656s + 1}e^{-0.8s}. \tag{14}$$

(3) An IBBO optimized iterative fractional-order transfer function model:

$$M(s) = \frac{0.9905}{38.0346s^{1.0708} + 1}e^{-0.8s}. \tag{15}$$

A comparison of the step response curves of the three models and the normalized data from the open-loop experiments of the pneumatic control valve is shown in Figure 4.

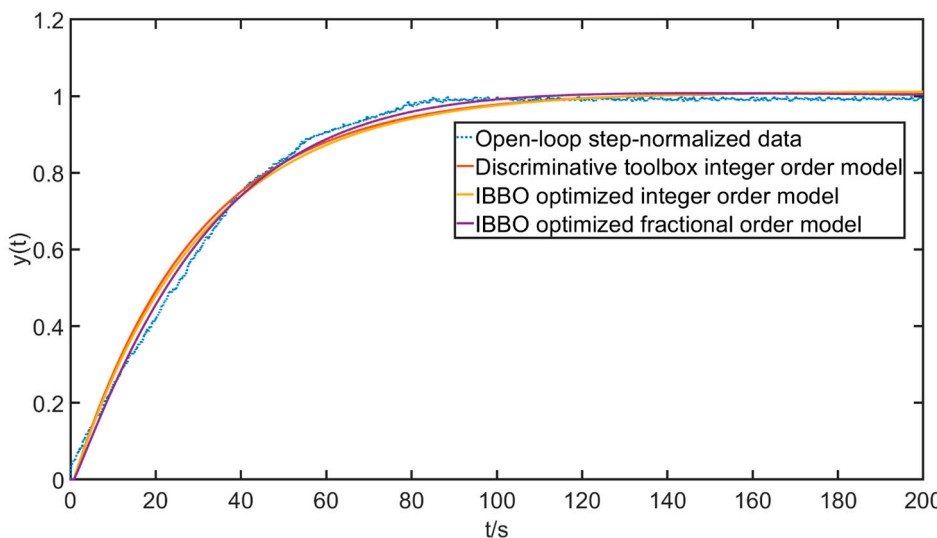

**Figure 4.** The plot of the step response curve against the original data.

To evaluate the deviation of the step responses of the three models concerning the actual open-loop experiment, the relative error is introduced:

$$\delta = \frac{(x - \mu)}{\mu} \times 100\%, \tag{16}$$

where $\delta$ is the actual relative error, generally given by a percentage, $x$ is the measured value (the simulation results of the three models respectively), and $\mu$ is the true value (open-loop step experiment results), as shown in Figure 5.

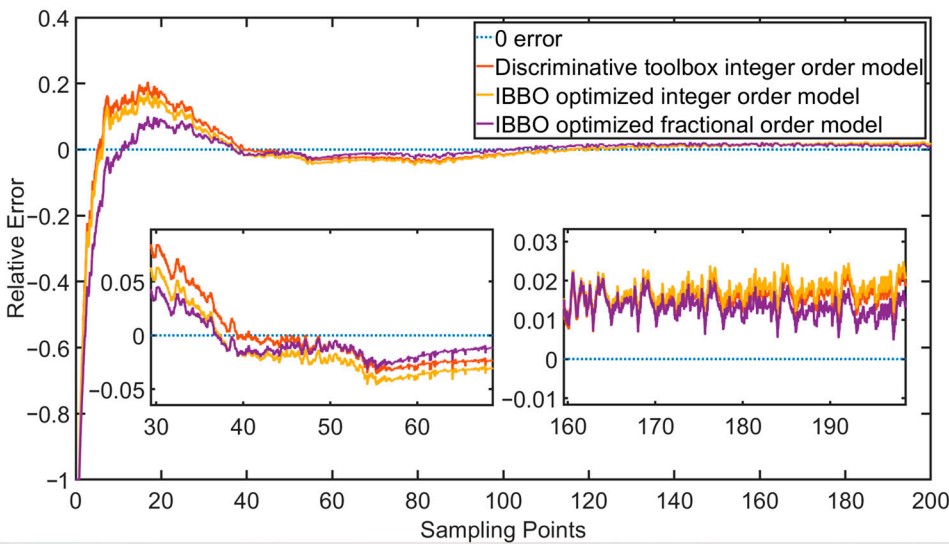

**Figure 5.** Relative error curves for the three models at each sampling point.

Additionally, the mean absolute error (*MAE*) is introduced here, and the respective *MAE* values of three models can be calculated by Equation (17):

$$MAE = \frac{1}{n}\sum_{i=1}^{n}|\hat{y}_i - y_i|, \tag{17}$$

where $n$ is the total number of sampling points, and $\hat{y}_i$ and $y_i$ are the predicted and actual values of the sampling point $i$, respectively. The final results are shown in Table 1.

**Table 1.** Errors in the step responses of three models with open-loop experiments.

| | MATLAB Recognition Toolbox Integer-Order Model | IBBO Optimized Iterative Integer-Order Model | IBBO Optimized Iterative Fractional-Order Model |
|---|---|---|---|
| $MAE = \frac{1}{n}\sum_{i=1}^{n}|\hat{y}_i - y_i|$ | 2.024% | 2.117% | 1.463% |
| $E = \sum_{i=1}^{n} e(i)^2$ | 3.347% | 3.137% | 1.416% |

Both the figure and the table show that the fractional-order model iterated via IBBO optimization has less errors and higher accuracy, which is more suitable for the actual pneumatic control valve operating conditions.

## 4. Controller Design

### 4.1. Integer-Order Controller Design

To design the controller for the valve positioner, the transfer function model of the inertia plus time lag of integer order iterated via IBBO optimization above, i.e., Equation (18), is selected for an analysis:

$$M(s) = \frac{1.0127}{29.9656s + 1}e^{-0.8s}. \tag{18}$$

The most widely used, convenient, and simple PID control algorithm in industrial production is selected for comparison and an analysis with internal mode control. The Z-N tuning method [28] mainly includes the Z-N step-up empirical method and the Z-N critical

proportion method, and for a model like Equation (8), its PID tuning formula is shown in Table 2.

**Table 2.** Z-N tuning method.

|                                | Kp        | Ti     | Td       |
| ------------------------------ | --------- | ------ | -------- |
| Z-N step-up empirical method   | $1.2T/KL$ | $2L$   | $0.5L$   |
| Z-N critical proportion method | $0.6Ku$   | $0.5Tu$| $0.125Tu$|

*Ku* and *Tu* are the critical gain and critical oscillation periods, respectively.

The transfer function of the PID control strategy obtained by the Z-N soaring empirical method takes the following form:

$$G(s) = 44.3847(1 + \frac{1}{1.6s} + 0.4s). \tag{19}$$

The transfer function of the PID control strategy obtained by the Z-N critical proportion method then takes the following form:

$$G(s) = 35.6046\left(1 + \frac{1}{1.6025s} + 0.4006s\right). \tag{20}$$

*4.2. Integer-Order Internal Mode Controller Design*

For the closed-loop control system shown in Figure 6 with an open-loop transfer function of $G_l(s) = C(s)P(s)$, there is an associated definition of sensitivity as shown in Equation (21):

$$S(s) = \frac{1}{1 + G_l(s)} = \frac{1}{1 + C(s)P(s)}. \tag{21}$$

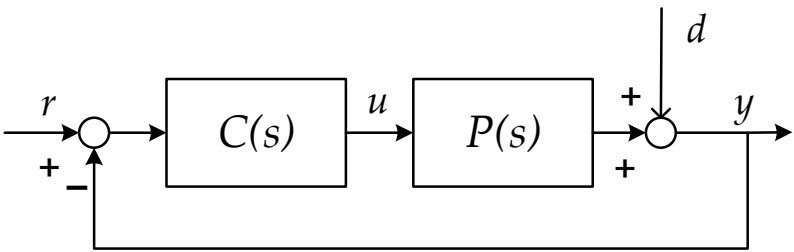

**Figure 6.** Closed-loop control systems.

The variation in the sensitivity performance comes from the adjustment of the process parameters by the closed-loop transfer function. If the system sensitivity is relatively high, this means that the stability of the system is very low, which is not conducive to the execution of the control system, and conversely the system robustness is high and can tolerate greater disturbance turbulence. We define the maximum magnitude of the sensitivity as the maximum sensitivity $M_S$, then we have:

$$M(s) = \max_{0 \leq \omega < \infty} |S(j\omega)| = \max_{0 \leq \omega < \infty} \left| \frac{1}{1 + C(j\omega)P(j\omega)} \right|. \tag{22}$$

The geometric interpretation is shown in Figure 7.

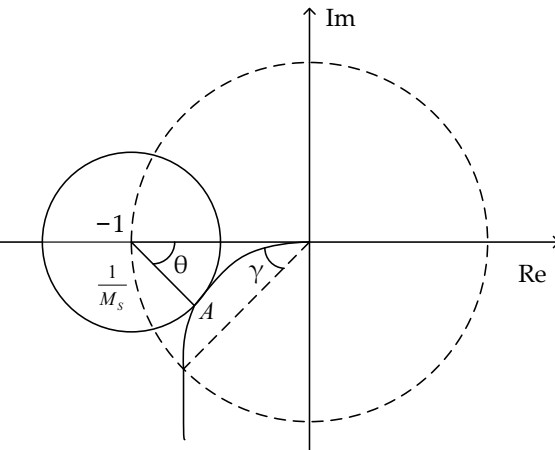

**Figure 7.** Geometric interpretation of maximum sensitivity.

$M_s$ is the inverse of the shortest distance from the Nyquist curve of the open-loop transfer function $C(j\omega)P(j\omega)$ to the critical point $(-1, j_0)$, i.e., the Nyquist curve of the open-loop transfer function is a tangent to the circle, with $M_S^{-1}$ as the radius and the critical point as the center. The relationship between the maximum sensitivity and the gain and phase margin indicators is as follows:

$$h_g > \frac{M_s}{M_s - 1},\tag{23}$$

$$\gamma > 2\arcsin\frac{1}{2M_S},\tag{24}$$

where $h_g$ and $\gamma$ are the gain and phase margin, respectively. Therefore, the maximum sensitivity can satisfy both gain and phase margin indexes, and the smaller the value of $M_S$, the larger the stability margin of the system. Typical $M_S$ values are in the range of 1.2–2.0, so the corresponding amplitude margin and phase angle margin are 6.0–2.0 and 49.2°–29.0°, respectively, and when $M_S = 1.5$, $h_g > 3.0$, and $\gamma > 38.9°$, it can be seen that when $M_S$ takes certain values, the corresponding magnitude margin and phase angle margin also satisfy certain conditions.

Thus, in Figure 7, assuming that point $A$ is the tangent point, the Nyquist curve of the open-loop transfer function $G_l(j\omega)$ traversing point $A$ is conditional on:

$$G_l(j\omega) = -1 + \frac{1}{M_S}e^{-j\theta},\tag{25}$$

$$\arg\frac{dG_l(j\omega)}{d\omega} = \frac{\pi}{2} - \theta.\tag{26}$$

where $\theta$ is the angle between the negative real axis and the line connecting the critical point and point $A$.

Assuming that the model of the established system is nondifferential, i.e., $P(s) = M(s)$, then the frequency characteristic of the open-loop transfer function of the system is obtained according to the feedback controller Equation (1) as:

$$G_l(j\omega) = C(j\omega)P(j\omega) = \frac{Q(j\omega)}{1 - Q(j\omega)M(j\omega)}P(j\omega),\tag{27}$$

$P(s) = M(s)$, Equation (27) can be reduced to:

$$G_l(j\omega) = \frac{Q(j\omega)}{1 - Q(j\omega)P(j\omega)}P(j\omega),\tag{28}$$

Substituting the IMC controller Equation (29) yields:

$$Q(s) = f(s)M_-^{-1}(s), \tag{29}$$

$$G_l(j\omega) = \frac{M_+(j\omega)f(j\omega)}{1 - M_+(j\omega)f(j\omega)}. \tag{30}$$

By substituting filter Equation (31) while taking the order of the filter as $\gamma_1 = 1$, $n = 1$, after finishing we get the following equation:

$$f(s) = \begin{pmatrix} \frac{1}{(1+\lambda_1 s)^{\gamma_1}} & & \\ & \ddots & \\ & & \frac{1}{(1+\lambda_n s)^{\gamma_n}} \end{pmatrix}, \tag{31}$$

$$G_l(j\omega) = \frac{e^{-j\omega L}}{1 + j\omega\lambda - e^{-j\omega L}}, \tag{32}$$

Combining Equations (25), (26), and (32), we have the following formula:

$$\begin{cases} \frac{1}{(\lambda+L)\omega}\sin(\omega L) = 1 - \frac{1}{M_S}\cos\theta \\ \frac{1}{(\lambda+L)\omega}\cos(\omega L) = \frac{1}{M_S}\sin\theta \\ \omega L + \arctan\frac{1}{\omega L} = \frac{\pi}{2} + \theta \end{cases}. \tag{33}$$

Equation (33) is a nonlinear equation with no direct solution. Using MATLAB to calculate and analyze the nonlinear relationship, the approximate approximation expression for the parameter $\lambda$ is obtained as:

$$\lambda = \frac{1.508 - 0.451 M_S}{1.451 M_S - 1.508} L. \tag{34}$$

According to Equation (34), by determining the value of the maximum sensitivity $M_S$, the corresponding value of the unique parameter $\lambda$ of the internal mode controller can be uniquely determined.

Thus, analyzing the model for Equation (18), associating Equations (1), (29), and (31), we have:

$$\begin{aligned} C(s) &= \frac{Q(s)}{1 - Q(s)M(s)} = \frac{f(s)M_-^{-1}(s)}{1 - f(s)M_-^{-1}(s)M(s)} \\ &= \frac{\frac{1}{\lambda s+1}\frac{Ts+1}{K}}{1 - \frac{1}{\lambda s+1}\frac{Ts+1}{K}\frac{K}{Ts+1}e^{-Ls}} \\ &= \frac{Ts+1}{K(\lambda s+1-e^{-Ls})} \approx \frac{Ts+1}{K(\lambda+L)s} \end{aligned} \tag{35}$$

Here, the low-pass filter is taken as $f(s) = 1/(\lambda s + 1)$, while the first-order Taylor approximation expansion $e^{-Ls} = 1 - Ls$ is used for the delay link, and when $M_S$ is taken as 1.6, $\lambda$ corresponds to a value of 0.7733, so that the corresponding internal mode control feedback controller is:

$$C(s) = \frac{29.9656s + 1}{1.5933s}, \tag{36}$$

Converting Equation (36) to the control form of the internal mode PID, we have:

$$C(s) = 18.8073\left(1 + \frac{1}{29.9657s}\right). \tag{37}$$

### 4.3. Fractional-Order Internal Mode Controller Design

As explained in the previous chapters, when $F(s) = I$, the equivalent feedback internal mode control structure shown in Figure 2 is plotted as a single-degree-of-freedom system, i.e., a one-degree-of-freedom system. Its convenience comes from the fact that there is only

one adjustable filtering parameter, which is quite friendly to researchers, but unfortunately it requires a compromise between the tracking and robustness of the control system and cannot achieve both a fish and bear's paw. In a later section, a feasible approach is proposed to solve this problem. Here, the concept of fractional-order PID is introduced to design a single-degree-of-freedom membrane controller.

Standard PID controllers are popular among researchers because of their adaptability, high robustness, and ease of implementation. However, it is difficult to achieve the target control requirements in the face of more complex environments. In recent years, as researchers have gained a better understanding of fractional-order PID, its advantages have become more and more evident [29]. The general format of fractional-order PID is $\text{PI}^\lambda\text{D}^\mu$. Since it has more differential and integral orders than the standard PID, the parameter adjustment range of the controller will become larger, and the control flexibility of the target object will increase [30], so the quality of control will also be improved.

Thus, the design of a single-degree-of-freedom internal mode control system for a fractional order of a type similar to Equation (38) is performed:

$$M(s) = \frac{0.9905}{38.0346s^{1.0708} + 1}e^{-0.8s},\tag{38}$$

$$M(s) = \frac{K}{Ts^\alpha + 1}e^{-Ls},\tag{39}$$

where $K$, $T$, and $L$ are positive real numbers and the value range of $\alpha$ is $0, 2$.

According to the above internal mode control design steps, Equation (38) is first decomposed into the unstable part and the minimum phase part:

$$M(s) = M_+(s)M_-(s).\tag{40}$$

However, before that the time lag part $e^{-Ls}$ needs to be equivalently approximated. Various scholars have proposed several methods to deal with it, among which the better known ones include the first-order Padé approximation expansion:

$$e^{-Ls} = \frac{1 - \frac{L}{2}s}{1 + \frac{L}{2}s},\tag{41}$$

and first-order Taylor approximation expansion:

$$e^{-Ls} = 1 - Ls.\tag{42}$$

To facilitate the calculation, the first-order Padé approximation is chosen here, which is decomposed according to Equation (40), with:

$$M_+(s) = 1 - \frac{L}{2}s,\tag{43}$$

$$M_-(s) = \frac{K}{(Ts^\alpha + 1)\left(1 + \frac{L}{2}s\right)}.\tag{44}$$

Thus, the fractional order internal mode controller (FOIMC) is obtained:

$$Q(s) = f(s)M_-^{-1}(s) = \frac{(Ts^\alpha + 1)\left(1 + \frac{L}{2}s\right)}{K(\lambda s + 1)}.\tag{45}$$

At this point, $f(s) = 1/(\lambda s + 1)$, from Equation (1), and the feedback controller $C(s)$ of the whole internal mode control system is:

$$C(s) = \frac{(Ts^\alpha + 1)\left(1 + \frac{L}{2}s\right)}{K\left(\lambda + \frac{L}{2}\right)s}, \tag{46}$$

translating Equation (46) into the form of a fractional order PID, there are three cases:

(1) When $0 < \alpha < 1$:

$$C(s) = \frac{1}{K\left(\lambda + \frac{L}{2}\right)}\left(\frac{L}{2} + \frac{1}{s} + \frac{T}{s^{1-\alpha}} + \frac{TL}{2}s^\alpha\right). \tag{47}$$

(2) The special case when $\alpha = 1$ is the standard integer-order PID model:

$$C(s) = \frac{1}{K\left(\lambda + \frac{L}{2}\right)}\left(T + \frac{L}{2} + \frac{1}{s} + \frac{TL}{2}s\right). \tag{48}$$

(3) When $1 < \alpha < 2$:

$$C(s) = \frac{1}{K\left(\lambda + \frac{L}{2}\right)}\left(\frac{L}{2} + \frac{1}{s} + Ts^{\alpha-1} + \frac{TL}{2}s^\alpha\right). \tag{49}$$

It can be seen that this FOPID contains integral terms of the integer order and fractional order $\alpha - 1$, in addition to proportional links and the differentiation of order $\alpha$.

For the smart valve positioner system model shown in Equation (38), from Equation (34), $M_S$ is taken to be 1.6 and $\lambda$ is taken to be corresponding, so that the corresponding internal mode control feedback controller is:

$$C(s) = 0.8605\left(0.4 + \frac{1}{s} + 38.0346s^{0.0708} + 15.2138s^{1.0708}\right). \tag{50}$$

If a similar design approach to the integer-order is used for the fractional-order model, there is a fractional-order feedback controller:

$$C(s) = \frac{38.0346s^{1.0708} + 1}{1.5584s}. \tag{51}$$

*4.4. Two-Degree-of-Freedom Fractional-Order Internal Mode Controller Design*

To balance the contradiction between the dynamic response characteristics and robust stability of the internal mode control system while meeting the control requirements and improving the control quality, a two-degree-of-freedom internal model control system is introduced, as shown in Figure 8:

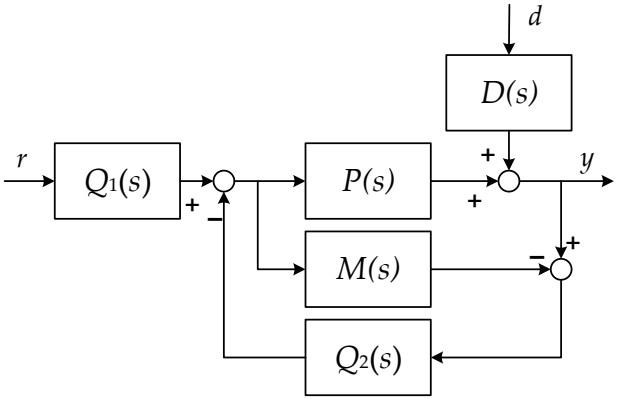

**Figure 8.** Diagram of a two-degree-of-freedom control system.

Similarly, this can be translated into the equivalent feedback two-degree-of-freedom control system structure diagram shown in Figure 9:

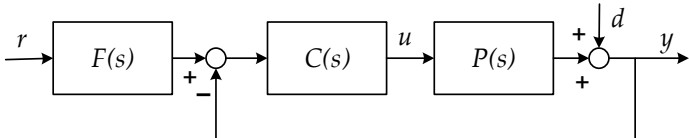

**Figure 9.** Equivalent feedback two-degree-of-freedom control system structure diagram.

In the figure, $Q_1(s)$ and $Q_2(s)$ are two-degree-of-freedom internal-mode controllers, $C(s)$ is the feedback controller of the two-degree-of-freedom control system, and $F(s)$ is the set value filter, where:

$$C(s) = \frac{Q_2(s)}{1 - M(s)Q_2(s)}, \tag{52}$$

$$F(s) = \frac{Q_1(s)}{Q_2(s)}. \tag{53}$$

Thus, under the premise that the model is free of differences, i.e., $P(s) = M(s)$, we have:

$$y = M(s)Q_1(s)r + [1 - M(s)Q_2(s)]d. \tag{54}$$

From Equation (49), it can be seen that $Q_1(s)$ can change the dynamic response performance of the system, while $Q_2(s)$ can be adjusted to suppress the external disturbance signal.

Thus, for Equation (34), the internal mode controllers $Q_1(s)$ and $Q_2(s)$ are set according to the principle design steps of internal mode control as follows:

$$\begin{cases} Q_1(s) = f_1(s)M_-^{-1}(s) \\ Q_2(s) = f_2(s)M_-^{-1}(s) \end{cases}, \tag{55}$$

We take the low-pass filters $f_1(s)$ and $f_2(s)$ as:

$$\begin{cases} f_1(s) = \frac{1}{1+\lambda_1 s} \\ f_2(s) = \frac{1}{1+\lambda_2 s} \end{cases}. \tag{56}$$

Thus, substituting Equations (39), (41), (44), (55) and (56) into Equation (52) yields the feedback controller for the two-degree-of-freedom control system:

$$C(s) = \frac{(Ts^\alpha + 1)\left(1 + \frac{L}{2}s\right)}{K\left(\lambda_2 + \frac{L}{2}\right)s}. \tag{57}$$

Additionally, substituting Equations (44), (55), and (56) into Equation (53) gives the setpoint filter of the two-degree-of-freedom control system as:

$$F(s) = \frac{1 + \lambda_2 s}{1 + \lambda_1 s}. \tag{58}$$

Again using the maximum sensitivity principle for the intelligent valve positioner system model shown in Equation (38), from Equation (34), taking $M_s$ as 1.6, $\lambda_2$ corresponds to a value of 0.7733, so that the corresponding feedback controller for the two-degree-of-freedom internal-mode control system is:

$$C(s) = 0.8605\left(0.4 + \frac{1}{s} + 38.0346s^{0.0708} + 15.2138s^{1.0708}\right), \tag{59}$$

$$F(s) = \frac{1 + 0.7733s}{1 + 1.0000s}. \tag{60}$$

There is no effective parameter adjustment method corresponding to the value of $\lambda_1$; however, in general, with the constant $\lambda_2$, as the value of $\lambda_1$ increases, the dynamic response performance of the system will become weaker but the amount of overshoot will be reduced accordingly. On the contrary, as the value of $\lambda_1$ decreases, the response speed of the system will be accelerated but the overshoot of the system will be increased. After several debugging and experiments, the value of $\lambda_1$ should be slightly larger than the value of $\lambda_2$, which will be reflected in a later section.

## 5. Simulation and Experimentation

For the integer-order pneumatic control valve model of Equation (14), there are three control methods, namely Equation (19), the fly-up empirical method, Equation (20), the critical proportional method, and Equation (37), the integer-order internal mode PI control, and the control flow simulation model is constructed in Simulink as shown in Figure 10:

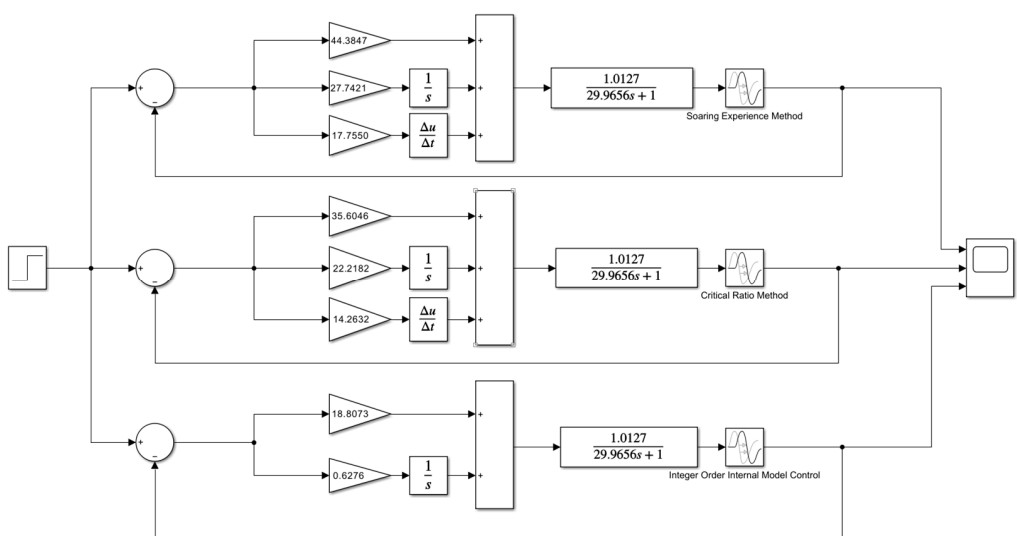

**Figure 10.** Simulink model for integer-order internal mode control.

Its step control response curve is shown in Figure 11.

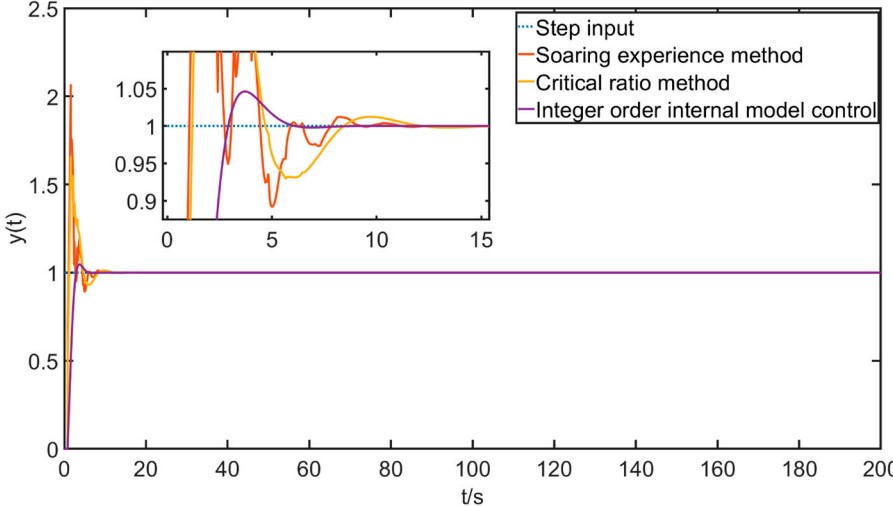

**Figure 11.** Step response integer-order control curve.

The dynamic performance response indicators are shown in Table 3.

**Table 3.** Dynamic performance indicators for integer-order control.

| Dynamic Performance Metrics | Overshoot | Rise Time | Peak Time | Adjustment Time |
| --- | --- | --- | --- | --- |
| Z-N soaring experience method | 106.464% | 0.171 s | 1.599 s | 5.544 s |
| Z-N critical ratio method | 65.655% | 0.301 s | 1.599 s | 7.071 s |
| Integer-order internal mode control | 4.612% | 1.468 s | 3.682 s | 2.648 s |

The relevant evaluation indicators are explained as follows. The overshoot is the percentage of the peak value exceeding the final value, the rise time is the time required for the step response to go from 10% to 90% of the final value, the peak time is the time for the step response to cross the final value and reach the first peak, and the adjustment time is the minimum time required for the step response to reach and remain within a 5% error band of the final value. From the chart together, it can be seen that although the Z-N critical proportion method has a better control effect and quality than the fly-up empirical method to some extent, it still fails to meet the demand for industrial control, while the integer-order internal mode PI control has a better effect.

Adding a perturbation error of −0.3 at 100 s and lasting 1 s to examine the robust stability of several control algorithms, there is the following as shown in Figure 12.

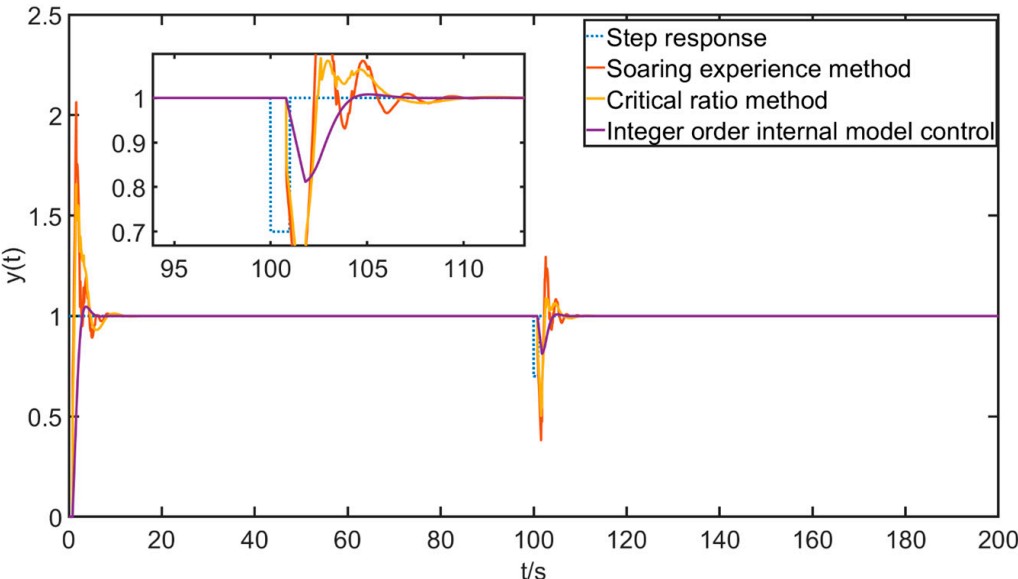

**Figure 12.** Step response perturbation error response curve.

It can be seen that the integer-order internal mode controller can regulate the output to a given value faster and has a better ability to suppress external disturbances.

The integer-order internal mode controller with the best control effect is compared with the fractional-order internal mode control with one degree of freedom from Equation (50) for the fractional-order pneumatic control valve model of Equation (15), the fractional-order internal mode PI control of Equation (51), and the two-degree-of-freedom fractional-order internal mode control of Equations (59) and (60) for the MATLAB simulation, and the Simulink simulation model is shown in Figure 13.

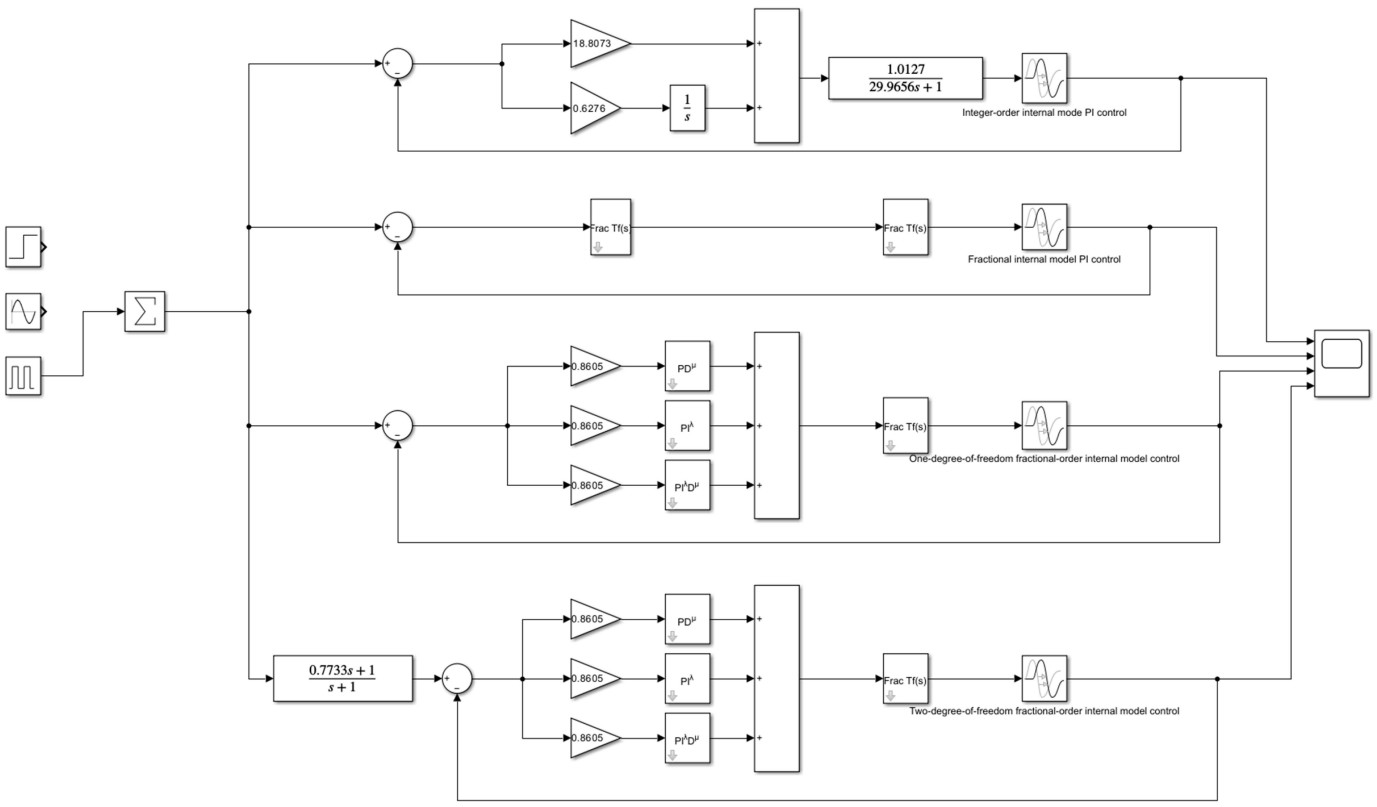

**Figure 13.** Simulink model for fractional-order internal mode control.

The step response curve is shown in Figure 14 below.

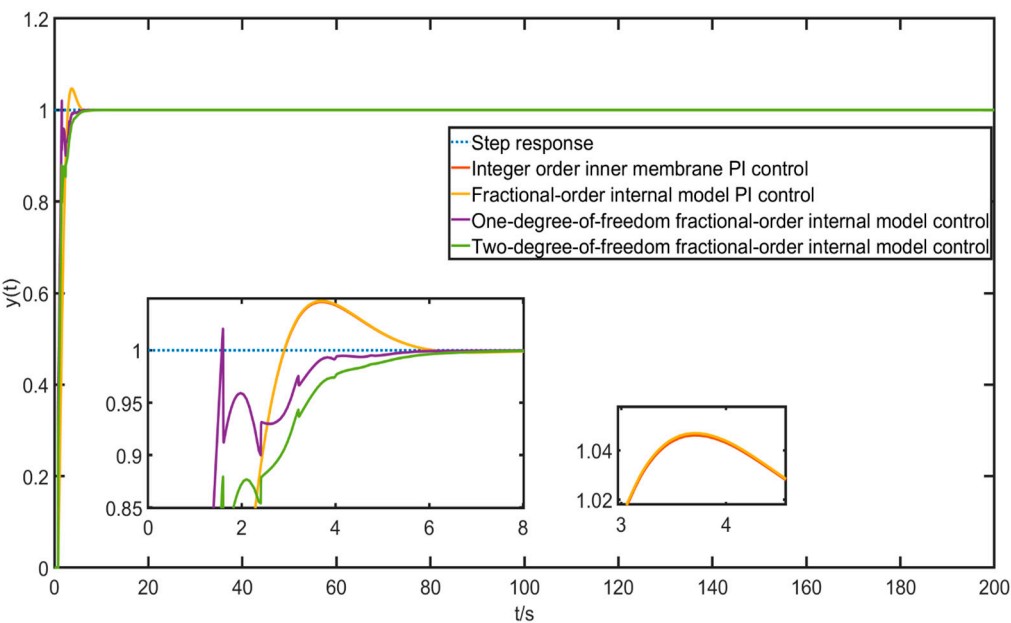

**Figure 14.** Step response fractional-order control curve.

The dynamic performance response indicators are shown in Table 4.

**Table 4.** Dynamic performance indicators for integer-order control.

| Dynamic Performance Metrics | Overshoot | Rise Time | Peak Time | Adjustment Time |
|---|---|---|---|---|
| Integer-order internal mode PI control | 4.6122% | 1.4675 s | 3.6824 s | 2.6482 s |
| Fractional-order internal mode PI control | 4.6969% | 1.4675 s | 3.6824 s | 2.6482 s |
| One-degree-of-freedom fractional-order internal mode control | 2.0561% | 0.5963 s | 1.5994 s | 3.0470 s |
| Two-degree-of-freedom fractional-order internal mode control | 0.0002% | 2.02415 s | 15.5554 s | 3.4121 s |

It can be seen here that the same method of a control algorithm for integer-order or fractional-order models, i.e., integer-order internal mode PI control and fractional-order internal mode PI control, has almost the same effect (they all have the advantages of small overshoot and fast speed), so the following sections will not compare fractional-order internal mode PI control methods, and integer-order internal mode PI control will be used instead.

In order to further analyze the differences between several control algorithms and to find the optimal control algorithm, given a sinusoidal signal as the desired signal of $y = 0.5\sin(\pi t/20) + 0.5$, the simulation results and tracking errors are obtained as shown in Figures 15 and 16 below.

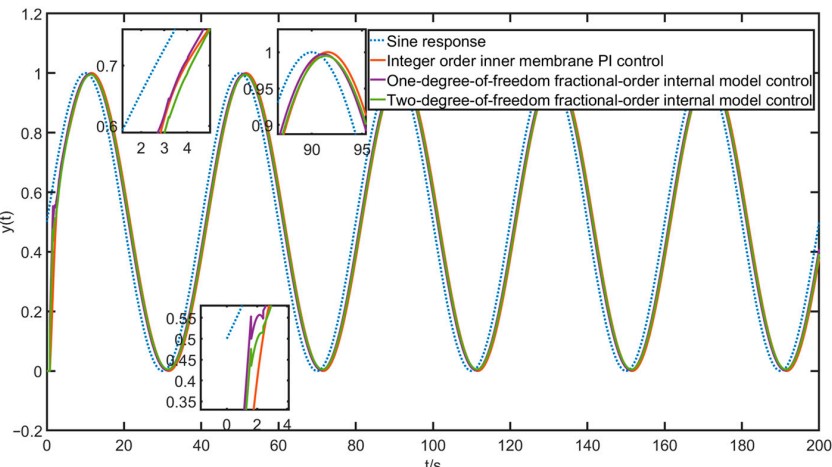

**Figure 15.** Sine simulation curve.

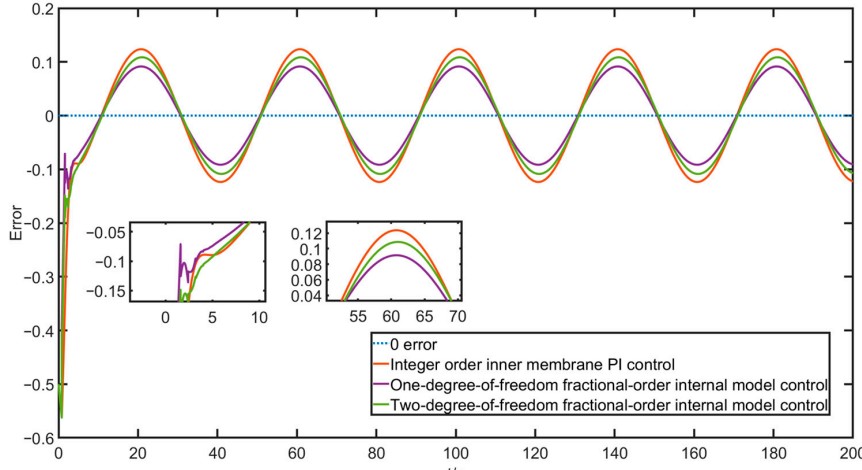

**Figure 16.** Sine tracking error.

As can be seen from the figure, although the rising curve of the integer-order internal mode PI control is smoother, the time used to converge to a steady state will be longer; that is, the tracking effect will be slightly delayed, while the tracking effect and error of the one-degree-of-freedom fractional-order internal mode control are better but the rising curve jitter in 0–2$s$ is more violent, more moderate, or the rise speed and tracking effect are available for the two-degree-of-freedom fractional-order internal mode control. Given a more severe square wave signal $y = 0.5 * square(\pi t/20 + \pi) + 0.5$, there are the following simulation results and tracking errors as shown in Figures 17 and 18.

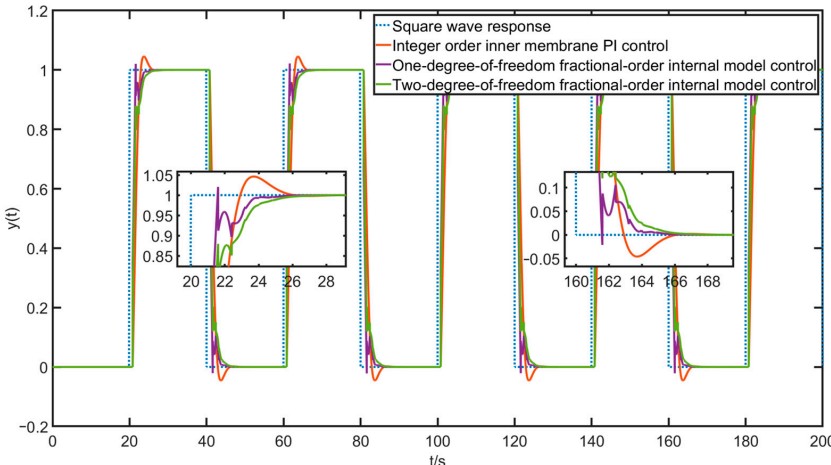

**Figure 17.** Square wave simulation curve.

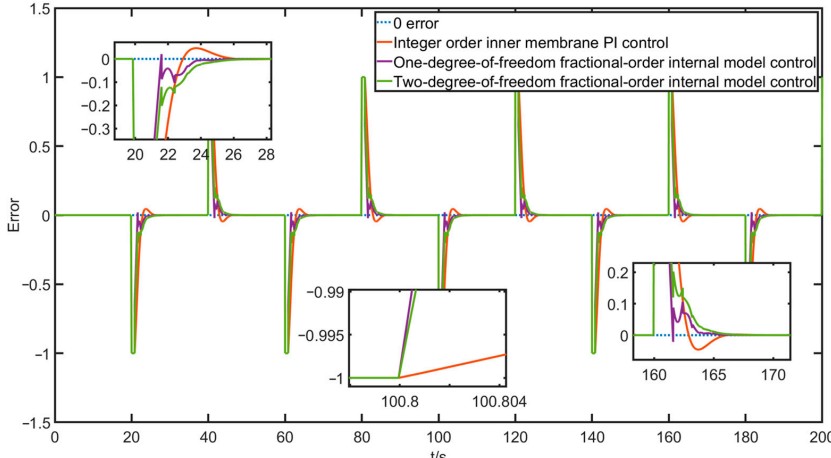

**Figure 18.** Square wave tracking error.

From the simulation curve of the square wave signal, the tracking error, and the above step and sine simulation experiments, the following conclusions can be further drawn. The overshoot of the two-degree-of-freedom fractional-order internal mode control algorithm is almost 0, and the rising curve is relatively smooth with almost no excess oscillation. Although the time used to reach the steady state is not as fast as the absolute radical one, it is acceptable within certain limits, which proves the effectiveness of the fractional-order model and the superiority of the two-degree-of-freedom fractional-order internal mode control.

The above simulation experiments, regarding the value of the maximum sensitivity $M_s$, are taken as the value $M_s = 1.6$, and according to Equation (34) we know that $\lambda = 0.7733$. The range of values of $M_s$ is 1.2–2.0, and the range of values is taken every 0.2. Simulation experiments are carried out with integer-order internal mode PI control algorithm, as shown in Figure 19 and Table 5.

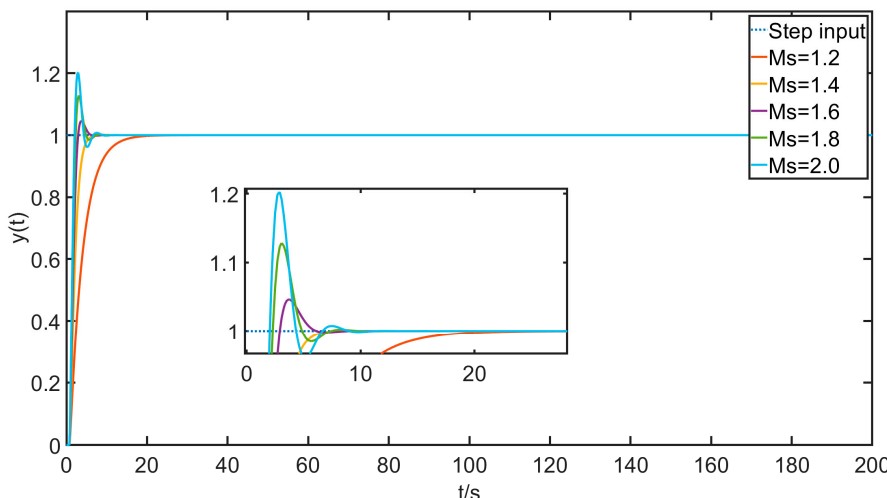

**Figure 19.** Integer-order control response curves for different $M_S$ with step inputs.

**Table 5.** Dynamic performance indicators for integer-order control.

| $M_S$ | $\lambda$ | Overshoot | Rise Time | Peak Time | Adjustment Time |
|-------|-----------|-----------|-----------|-----------|-----------------|
| 1.2 | 3.3166 | 0.0003% | 6.8669 s | 63.7906 s | 10.7269 s |
| 1.4 | 1.3366 | 0.0028% | 2.4496 s | 12.9635 s | 4.2802 s |
| 1.6 | 0.7733 | 4.6299% | 1.3033 s | 3.7537 s | 2.7006 s |
| 1.8 | 0.5046 | 12.8359% | 1.0196 s | 3.0713 s | 4.2802 s |
| 2.0 | 0.3478 | 20.2549% | 0.8080 s | 2.9057 s | 4.0645 s |

After analyzing the data shown in the above figure and table, it is clear that the overshoot increases dramatically with the increase in $M_S$, while the time required to reach the steady state or the dynamic response speed becomes better and better, which is a matter of trade-off and requires the researcher to make the right choice of $M_S$ for the control demand and quality in the actual situation.

The above simulation experiments on two-degree-of-freedom fractional-order internal mode control have been kept at the value of 1 for $\lambda_1$, and now the simulation experiments are conducted with different values of $\lambda_1$ while maintaining $M_S = 1.6$ and $\lambda_2 = 0.7733$, with the results shown in Figure 20 and Table 6 below.

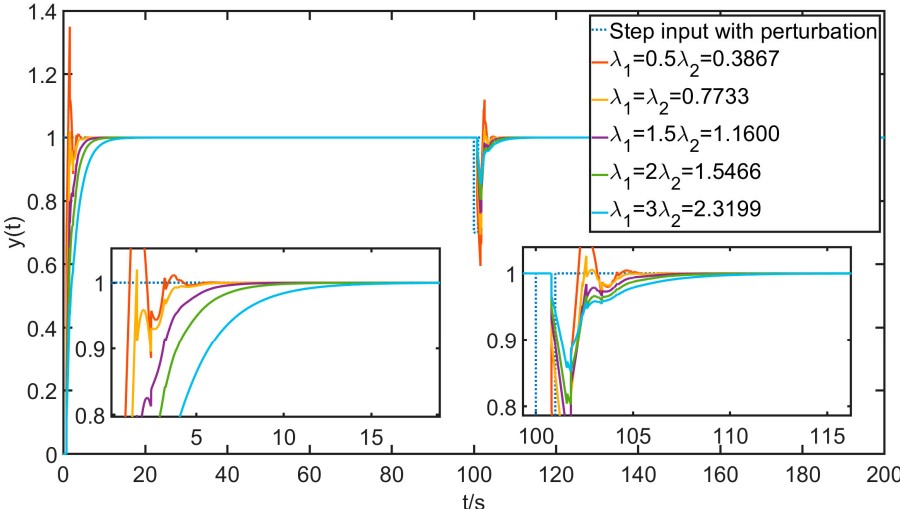

**Figure 20.** Response curve of two degrees of freedom fractional internal model control with different a values and perturbations under step input.

**Table 6.** Dynamic performance indicators for two-degree-of-freedom fractional-order internal mode control with different values of $\lambda_1$.

| $\lambda_1$ | Overshoot | Rise Time | Peak Time | Adjustment Time |
|---|---|---|---|---|
| $\lambda_1 = \dfrac{\lambda_2}{2} = 0.3867$ | 34.9644% | 0.2288 s | 1.5970 s | 2.8930 s |
| $\lambda_1 = \lambda_2 = 0.7733$ | 1.9810% | 0.6399 s | 1.6022 s | 3.0829 s |
| $\lambda_1 = 1.5\lambda_2 = 1.1600$ | 0.0033% | 2.2801 s | 18.1534 s | 3.7836 s |
| $\lambda_1 = 2\lambda_2 = 1.5466$ | 0.0008% | 2.9804 s | 122.4056 s | 5.0594 s |
| $\lambda_1 = 3\lambda_2 = 2.3199$ | 0.0010% | 4.8913 s | 93.1526 s | 7.4498 s |

As can be seen from the graphs and tables, when $\lambda_1 < \lambda_2$, the two-degree-of-freedom internal mode control will be out of tune, failing to meet the control requirements and becoming even worse than the general one-degree-of-freedom internal mode control, which has no research value, while when $\lambda_1 = \lambda_2$, this two-degree-of-freedom internal mode control is equivalent to the one-degree-of-freedom internal mode control. Finally, as for the case of $\lambda_1 > \lambda_2$, the control effect is significantly improved; as $\lambda_1$ increases, the rising curve becomes smoother and more resistant to disturbances, but the adjustment time required to reach the 5% error band increases significantly. Therefore, it is also necessary for the researcher to take the appropriate value of $\lambda_1$ for the control needs and quality in the actual situation.

In the MATLAB simulation in the above subsection, the superiority of the two-degree-of-freedom fractional-order internal mode control algorithm has been proved to a certain extent. To further verify this algorithm, the LabVIEW semi-physical simulation platform built in the laboratory will be optimized, and the pneumatic control valve system programs of integer-order internal mode PI control and two-degree-of-freedom fractional-order internal mode control will be built and experimentally analyzed, respectively.

The hardware and software semi-physical platform is set up for experimental research, and the step, sine, and square wave signals are set as the predetermined expected inputs of the valve level in the upper computer LabVIEW, while the above two control algorithms are experimented with. The experimental environment is described as follows. The regulating valve model is Lotte Autocontrol ZJHP and the temperature is 25 degrees Celsius, with normal atmospheric pressure. A physical picture of the experimental platform is shown in Figure 21.

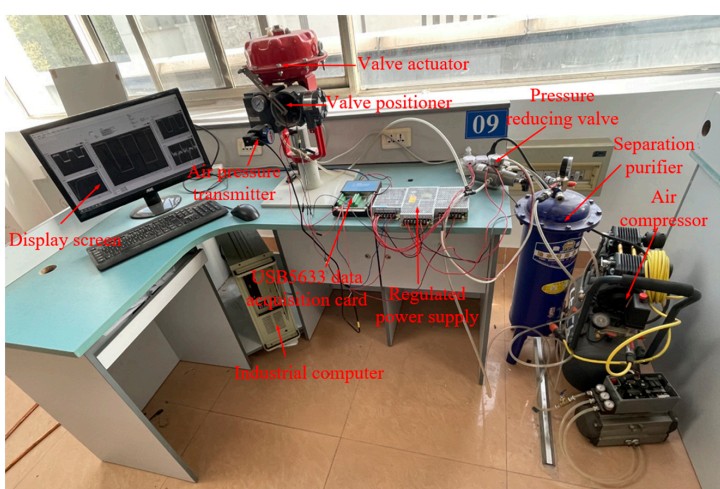

**Figure 21.** Experimental equipment and devices.

A block diagram of the system control operation flow of the LabVIEW-based semi-physical simulation experiment is shown in Figure 22 below.

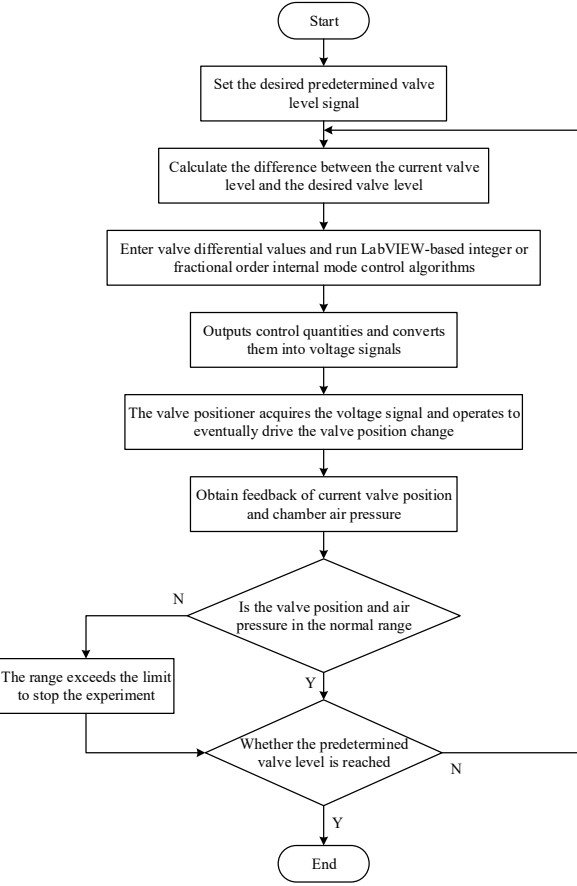

**Figure 22.** Flowchart of the LabVIEW-based experimental control system.

Firstly, the experiment on the step signal is conducted to mainly test the transient response performance of the control algorithm, given the desired input of 50% valve position opening. After the experiment, the valve position opening data collected by the data acquisition card are recorded with the gas chamber air pressure data and the response curve is generated in MATLAB. The results of the step experiment are shown in Figures 23–25.

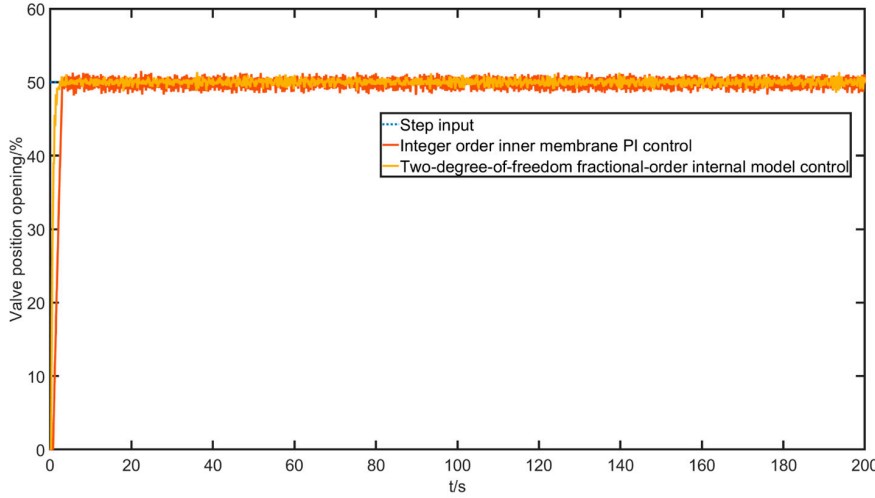

**Figure 23.** Valve position opening response curve with a step input.

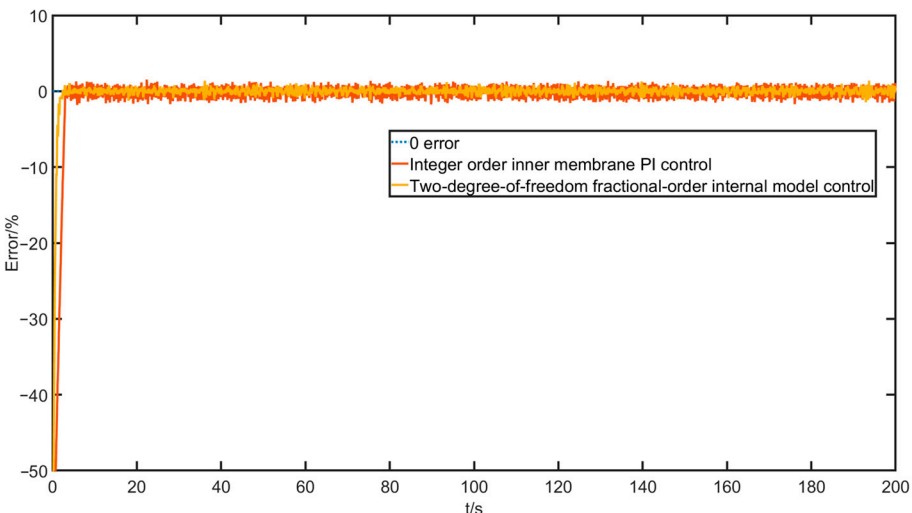

**Figure 24.** Valve position opening error curve with a step input.

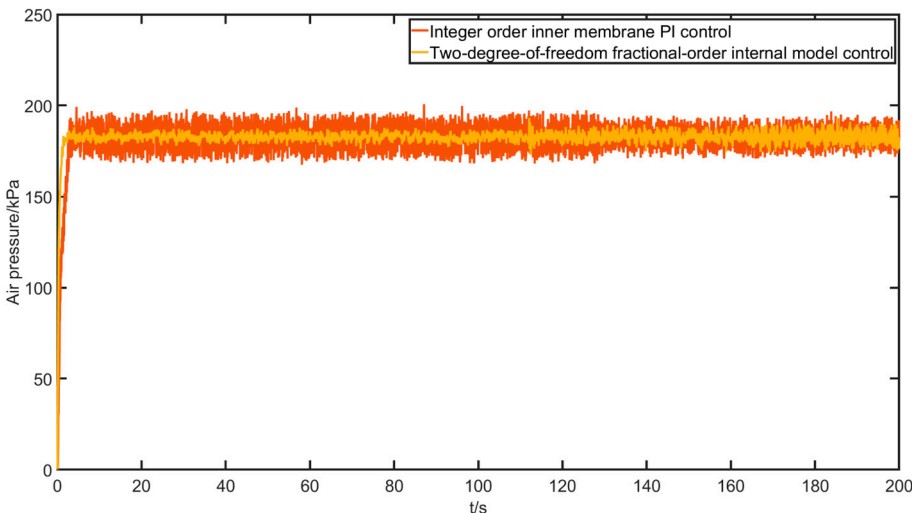

**Figure 25.** Chamber pressure response curve with a step input.

The performance indicators are shown in Table 7.

**Table 7.** Evaluation indicators for valve position opening control with a step input.

| Dynamic Performance Metrics | Overshoot | Rise Time | Peak Time | Adjustment Time | RMSE | MAPE |
|---|---|---|---|---|---|---|
| Integer-order internal mode PI control | 3.181% | 1.750 s | 3.050 s | 2.900 s | 4.299 | 1.998% |
| Two-degree-of-freedom fractional-order internal mode control | 2.795% | 0.750 s | 2.400 s | 1.650 s | 2.439 | 0.909% |

RMSE (root mean squared error) and MAPE (mean absolute percentage error) metrics are introduced here to measure the control quality. It can be seen that the two-degree-of-freedom fractional-order internal model control algorithm has significant advantages in terms of overshoot and speed. Then, a sinusoidal signal input experiment is conducted to test the dynamic performance and following characteristics of the control algorithm, given a desired valve opening signal with the shape $y = 30\sin(\pi t/20) + 50$. The experimental results are shown in Figures 26–28.

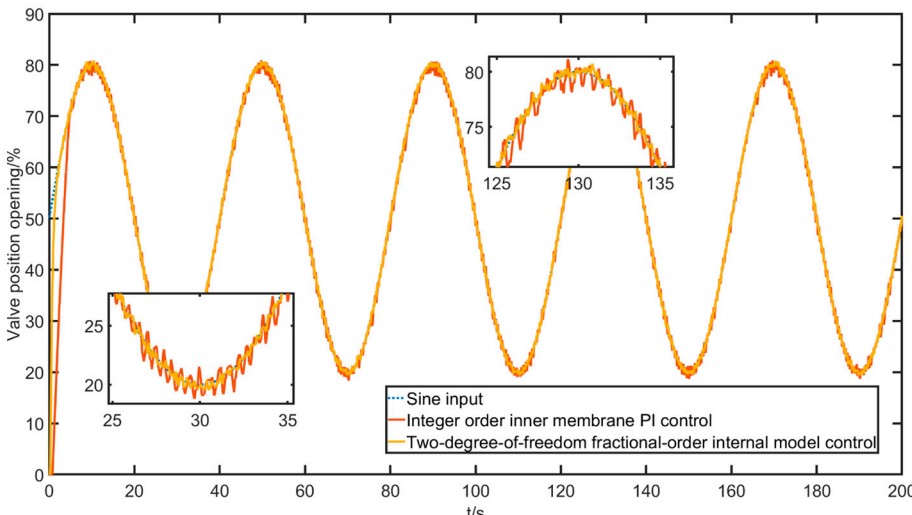

**Figure 26.** Valve position opening response curve with a sinusoidal input.

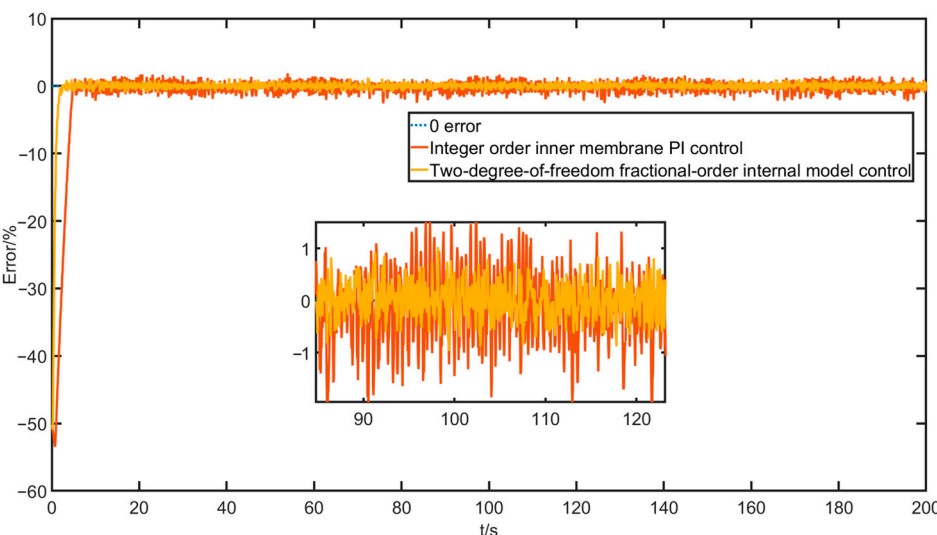

**Figure 27.** Valve position opening error curve with sinusoidal input.

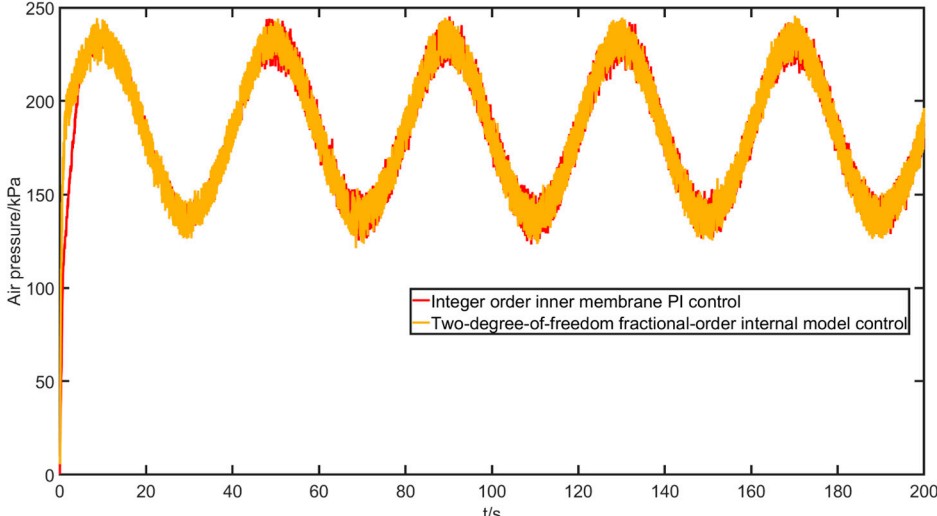

**Figure 28.** Chamber pressure response curve with a sinusoidal input.

The deviation indicators are shown in Table 8.

**Table 8.** Indicator of valve position opening deviation with a sinusoidal input.

| Deviation Indicators | RMSE | MAPE |
|---|---|---|
| Integer-order internal mode PI control | 5.283 | 2.664% |
| Two-degree-of-freedom fractional-order internal mode control | 2.612 | 1.068% |

The sine wave experiment also shows that the oscillation of the two-degree-of-freedom fractional internal model control is well maintained and the error is also small.

Finally, a square wave signal input experiment is conducted to test the fast dynamic performance of the control algorithm and its ability to follow the abrupt signal, given a desired valve opening signal with the shape $y = 30\mathrm{square}(\pi t/20) + 50$, and the experimental results are shown in Figures 29–31.

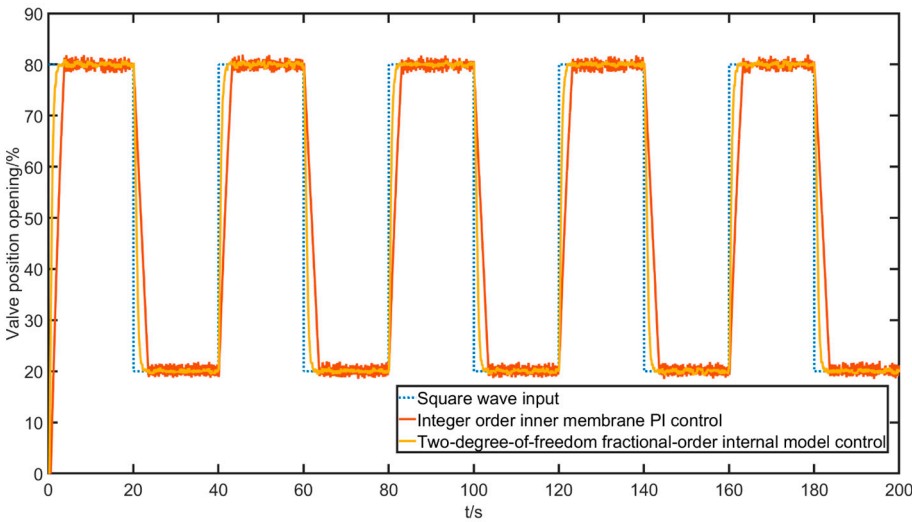

**Figure 29.** Valve position opening response curve with a square wave input.

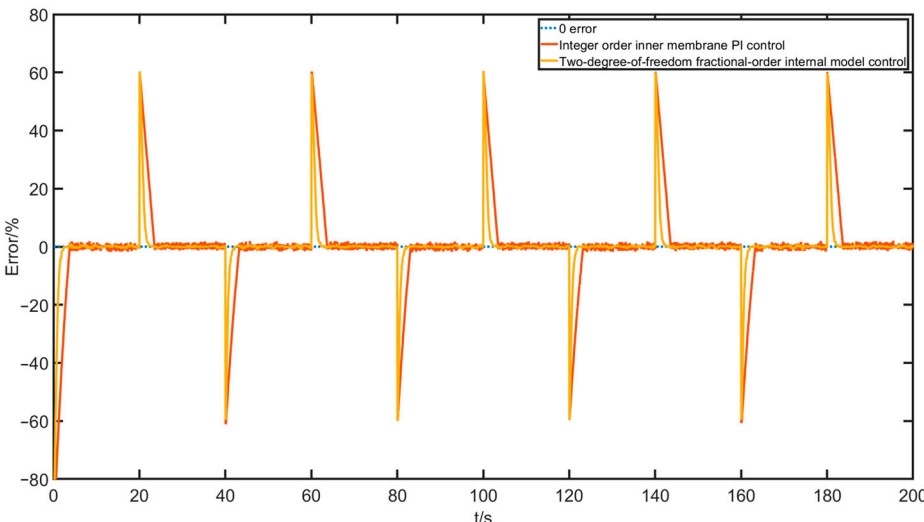

**Figure 30.** Valve position opening error curve with a square wave input.

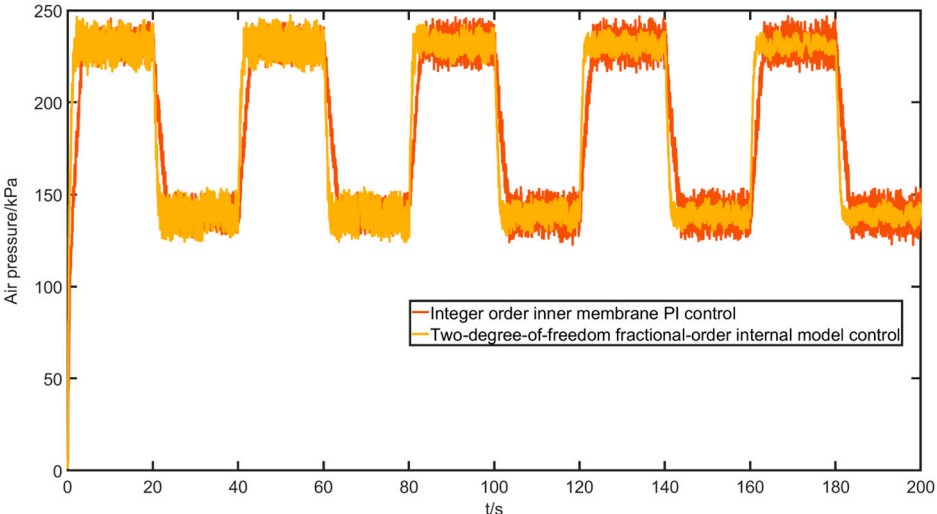

**Figure 31.** Chamber pressure response curve with a square wave input.

The deviation indicators are shown in Table 9.

**Table 9.** Indicator of valve position opening deviation with a square wave input.

| Deviation Indicators | RMSE | MAPE |
|---|---|---|
| Integer-order internal mode PI control | 15.212 | 18.666% |
| Two-degree-of-freedom fractional-order internal mode control | 9.343 | 7.571% |

According to the comparison of the data and indexes in the above figures and tables, it can be seen that the two-degree-of-freedom fractional-order internal mode control algorithm has a faster response, higher control accuracy, better tracking, and better response to abruptly changing signals than the integer-order internal mode PI control, i.e., the overall control quality is better.

## 6. Conclusions

This paper identified the pneumatic control valve by improving the biogeographic optimization algorithm, established the mathematical model of the pneumatic control valve, and on this basis proposed the valve position control method of the pneumatic control valve based on fractional-order two-degree-of-freedom internal mode control. The superiority of the proposed method was proven via simulation. Finally, experiments on pneumatic control valve position tracking control were carried out, the control programs of the two algorithms were written separately using Labview graphical programming software on the host computer, and the adaptability and effectiveness of the fractional-order algorithm and internal mode control in the pneumatic control valve position control system were verified according to the experiments. Through an experimental comparison, the two-degree-of-freedom fractional-order internal mode control achieved better dynamic performance than the integer-order internal mode PI control. Specifically, this control method improved the overshoot by 12%, rise time by 57%, peak time by 21%, regulation time by 43%, and RMSE and MAPE metrics by 43% and 57%, respectively, providing a reliable method for the control of the control valve position. In summary of the experiments, we can see that the fractional-order model has good adaptability and effectiveness in the field of pneumatic control valves, and the two-degree-of-freedom fractional-order internal mode control algorithm also effectively improves the accuracy, speed, and robustness performance of the valve position control.

**Author Contributions:** Conceptualization, M.Z.; methodology, M.Z. and S.C.; software, Z.X. and X.D.; validation, S.C. and Z.X.; formal analysis, S.C. and Z.X.; investigation, M.Z. and S.C.; resources, M.Z.; data curation, S.C. and Z.X.; writing—original draft preparation, S.C. and Z.X.; writing review and editing, M.Z. and X.D.; visualization, Z.X. and X.D.; supervision, M.Z.; project administration, M.Z.; funding acquisition, M.Z. All authors have read and agreed to the published version of the manuscript.

**Funding:** This research was funded by National Natural Science Foundation of China (62073113).

**Institutional Review Board Statement:** Not applicable.

**Informed Consent Statement:** Not applicable.

**Data Availability Statement:** Not applicable.

**Acknowledgments:** The authors gratefully appreciate the anonymous reviewers for their valuable comments.

**Conflicts of Interest:** The authors declare no conflict of interest.

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
