# Peer review of "Design of Two-Degree-of-Freedom Fractional-Order Internal Model Control Algorithm for Pneumatic Control Valves"

_actuators, doi:10.3390/act12050214_

Round 1

Reviewer 1 Report

We don't think the references cited in this paper are sufficient.

In Figure 21, the letters protrude from the diamond.

There are several types of pneumatic control valves, but we do not know which control valve the proposed method is effective for. That is, in this paper, some mathematical models of pneumatic control valves are shown, but we do not know how well they match the existing pneumatic control valves. In addition, it is necessary to actually conduct experiments using existing pneumatic control valves and evaluate control performance.

Correction by a native English speaker may be required.

Reviewer 2 Report

The paper describes a method for controlling a pneumatic valve. First a model of the  valve is generated with the "Biogeography-Based Optimization" algorithme. After obtaining the model, different control approaches are studied and investigated regarding raising time and overshoot. A fractional order approache is introduced, that can be seen as the noval core of this paper.  At the end the control concept is evaluated at an semi-phsyical simulation experiment.

The approach obtaining the model of the valve and the control is described in a good way. For the semi-physical simulation experiment the desciption of the test rig should be extended, in a way that the reader can better understand and evaluate the results. 

In the view of the reviewer a short introduction into fractional control seems to be usefull, because many readers are not familiar with this concept.

The figures with the simulink structures have a poor quality, it is necessary to improve the resolution. The figures with matlab plots have a better quality, nethertheless they can also improve in quality.

Figure(3):
Please put the text "Whether termination conditions are met" inside of the diamond

Equation(7):
In the opinion of the reviewer the equation should look like follows:

z=P(s)u+D(s)d according to figure 2 without considering M

Equation(8):
What kind of data is used to model the transfer function. The transfer function is not motivated with measurements, it is just written that is obtained with matlab recognition toolbox. Please give more details.

Equation(10):
Please give the model with this parameters e.g. M(s)=K/(Ts+1)exp(-Ls)=1.0127/(29.9656s+1)exp(-0.8s) 

Equation(49):
Please give the source for the fractional PID Controller. There are several definitions for it.

All equations: Please look at all equations and decide if the "," is the right sign or you should use a ".", when the equation is at the end of a sentance.

Line 91:
Remove the "Professor"

Line 176:
"Matlab recognition toolbox recognition", one recognition is to much

Line 196:
Here the reviewer would advises to give a short introduction to fractional order control

Line 202:
Put the heading on the next side

Line 215f:
what is the true value. The assumption of the reviewer is that x is the model value and my is the measured value??? Please claerify this.

Line 234: 
Z-N is Ziegler-Nichols?
Please give equations for obtaining K, Td, Ti

Line 274:
"n=1" in same line as the ","

Line 299:
"(0,2)" in same line with "."

Thank you for the interessting paper.

Reviewer 3 Report

Dear Authors,

This manuscript seems to be a well-structured, quality and scientific work.

Nevertheless, minor changes need to be done. 

Please find attached a detailed review.

Reviewer 4 Report

Research on Two-degree-of-freedom Fractional-order Internal Mode Control Algorithm Based on Pneumatic Control Valve
by Min Zhu, Siyuan Chen, Zihao Xu and Xueping Dong

Please, consider the following comments/remarks.

The title of the manuscript must be improved such that it conveys clearly what is the contribution of the work. The current title starts with the word "Research" ("Research on ..."). This makes the title too vague. From reading the manuscript, the work involves the testing of different approaches to design a controller to control a pneumatic valve. There is a typo along the paper in the designation of the control design method. The design method is based in an Internal Model Control (IMC) approach. Change the word "mode" to "model".

The abstract needs to be improved. It is necessary to clarify what is novelty that is highlighted in the work. That is, to state in an assertive manner what is/are the main contributions in the field of the control design for pneumatic control valves.

lines 14-16 (Abstract): "The superiority of the internal mode control algorithm is introduced and optimized, ..." The goal is not to introduce and to optimize the superiority of the internal model control algorithm. The sentence needs to be improved.

line 44: put in between parenthesis what PCM stands for.

Section 1: the organization of the review of the literature could be improved in terms of the different subjects that are addressed. Also, it is recommendable to rewrite the text with shorter sentences. This applies as well to other parts of the manuscript with too long sentences.

lines 80-81 and Section 2.1: it is not clear why the biogeography-based optimization (BBO) algorithm is more advantageous with respect with other optimization algorithms to calibrate the dynamic model of the pneumatic control valve. The description of the BBO algorithm can be shortened and adapted to the case study under consideration.

lines 137-138: check reference [20] with respect the authorship of the formulation of the Internal Model Control. If the authors are referring to the Internal Model Control (IMC) PID controller design, according to the literature this model-based design method was developed by Morari and coworkers (Garcia and Morari, 1982; Rivera et al. 1986).

Section 3: in the introduction of Section 3 (lines 166-176), cite the literature sources. The intelligent algorithm is the BBO algorithm.

lines 204-205: mention the algorithm/method used with the Matlab identification toolbox.

lines 360-364: these are the 3 "control methods" that were tested. Clarify the motivation to select these methods.

Section 5: the presentation and discussion of results can be improved, addressing each control design in separated subsections.

Section 6: It is not clear what were the improvements made to the BBO algorithm. The conclusions need to be clarified and completed.  Namely,
the conclusions (as well as the abstract)  must include  clear statements of what are the main contributions of the work.

It is difficult to follow in the paper the line of thinking in terms of the work regarding the pneumatic control valve model development and in terms of the model based control design method to control the valve.

Whenever possible rewrite the text with shorter sentences.

Round 2

Reviewer 4 Report

Article: Design of Two-degree-of-freedom Fractional-order Internal Model Control Algorithm for Pneumatic Control Valves
Authors: Min Zhu, Siyuan Chen, Zihao Xu and Xueping Dong

The article describes  the design of a two degree of freedom fractional order internal model control algorithm for pneumatic control  valves.  The methodology is applied to an experimental pneumatic control valve system. Therefore the contribution of the work is to demonstrate  an experimental application of PID control of a pneumatic valve, where the PID controller is designed using a two degree of freedom fractional order IMC approach. To do so, the dynamic model is calibrated using an optimization algorithm. Here, the biogeography based optimization (BBO) algorithm is used.  

1) The novelty/contributions of this work with respect to the previous work in reference 19 must be better highlighted in the introduction section. 

2) The abstract mentions that improvement methods are proposed. Here, to
be more specific about what are the improvement methods can help to
improve the abstract.

3) Provide a few references when mentioning that the biogeography based
optimization (BBO) algorithm has "received a lot of attention from scholars".

4) Explain better why the BBO algorithm is more advantageous in the
context of this particular application.

5) Increase the size of text in Figure 21.

6) Check for typos along the text. For example, see the text from line 288 to l. 297. Also, rewrite the text with shorter sentences.

7) In the l. 300: "\alpha is (0,2)". Remove the parentheses.

8) Change "Pade" to "Padé" .

Moderate editing of English language.
